# Characterization and Optimization of Multiomic Single-Cell Epigenomic Profiling

**DOI:** 10.3390/genes14061245

**Published:** 2023-06-10

**Authors:** Leticia Sandoval, Wazim Mohammed Ismail, Amelia Mazzone, Mihai Dumbrava, Jenna Fernandez, Amik Munankarmy, Terra Lasho, Moritz Binder, Vernadette Simon, Kwan Hyun Kim, Nicholas Chia, Jeong-Heon Lee, S. John Weroha, Mrinal Patnaik, Alexandre Gaspar-Maia

**Affiliations:** 1Division of Experimental Pathology, Department of Laboratory Medicine and Pathology, Mayo Clinic, Rochester, MN 55905, USA; assadmaiasandoval.leticia@mayo.edu (L.S.); mohammedismail.wazim@mayo.edu (W.M.I.); mazzone.amelia@mayo.edu (A.M.); dumbrava.mihai@mayo.edu (M.D.); munankarmy.amik@mayo.edu (A.M.); lee.jeongheon@mayo.edu (J.-H.L.); 2Epigenomics Program, Center for Individualized Medicine, Mayo Clinic, Rochester, MN 55905, USA; binder.moritz@mayo.edu (M.B.); kwanhyun58@gmail.com (K.H.K.); patnaik.mrinal@mayo.edu (M.P.); 3Mayo Clinic Medical Scientist Training Program, Mayo Clinic Alix School of Medicine and Mayo Clinic Graduate School of Biomedical Sciences, Mayo Clinic, Rochester, MN 55905, USA; 4Division of Hematology, Department of Internal Medicine, Mayo Clinic, Rochester, MN 55905, USA; fernandez.jenna@mayo.edu (J.F.); lasho.terra@mayo.edu (T.L.); 5Medical Genome Facility, Genome Analysis Core, Mayo Clinic, Rochester, MN 55905, USA; simon.vernadette@mayo.edu; 6Microbiome Program, Center for Individualized Medicine, Mayo Clinic, Rochester, MN 55905, USA; chia.nicholas@mayo.edu; 7Department of Medical Oncology, Mayo Clinic, Rochester, MN 55905, USA; weroha.saravut@mayo.edu

**Keywords:** single-cell sequencing, epigenomic profiling, snATAC-seq, snRNA-seq, nuclei preparation

## Abstract

The snATAC + snRNA platform allows epigenomic profiling of open chromatin and gene expression with single-cell resolution. The most critical assay step is to isolate high-quality nuclei to proceed with droplet-base single nuclei isolation and barcoding. With the increasing popularity of multiomic profiling in various fields, there is a need for optimized and reliable nuclei isolation methods, mainly for human tissue samples. Herein we compared different nuclei isolation methods for cell suspensions, such as peripheral blood mononuclear cells (PBMC, *n* = 18) and a solid tumor type, ovarian cancer (OC, *n* = 18), derived from debulking surgery. Nuclei morphology and sequencing output parameters were used to evaluate the quality of preparation. Our results show that NP-40 detergent-based nuclei isolation yields better sequencing results than collagenase tissue dissociation for OC, significantly impacting cell type identification and analysis. Given the utility of applying such techniques to frozen samples, we also tested frozen preparation and digestion (*n* = 6). A paired comparison between frozen and fresh samples validated the quality of both specimens. Finally, we demonstrate the reproducibility of scRNA and snATAC + snRNA platform, by comparing the gene expression profiling of PBMC. Our results highlight how the choice of nuclei isolation methods is critical for obtaining quality data in multiomic assays. It also shows that the measurement of expression between scRNA and snRNA is comparable and effective for cell type identification.

## 1. Introduction

Single-cell (SC) biology is a rapidly evolving field that interrogates various omics layers (genomics, epigenomics, transcriptomics, proteomics) (Figure 1) at SC resolution to define cellular heterogeneity in homeostasis and disease [1,2,3,4,5,6,7,8]. The correlation between epigenetic and genetic information, along with the analysis of phenotype through proteomic analysis, plays a critical role in uncovering the complex associations between molecular variations and observable traits. Integrating these diverse layers of molecular data allows for a comprehensive analysis of the interplay between genotype [9] and phenotype [10,11,12]. Consequently, the development of novel tools that enhance the precision and output of genotyping and phenotyping techniques holds substantial importance.

In order to ensure reliable results that provide the resolution and quality necessary to make meaningful scientific conclusions and to improve reproducibility, experimental protocols need to be optimized, evaluated, and standardized. This proves to be a challenge because several factors could affect the quality of the analyzed samples, leading to poor data quality. These include the type of tissue (PBMC, tumor, bone marrow, or cell line) which can impact the nuclear isolation protocol; the biological condition being studied (homeostasis, development, cancer, infection, or inflammation); the method of sample collection, handling, storage, and maintenance; cell viability and the number of cells available. Moreover, most protocols are typically defined using ideal samples that do not reflect the clinical and translational research environment, the so-called “real world” samples [12].

Epigenomic profiling of DNA accessibility (open chromatin regions) helps create a genome-wide characterization of transcription factor (TF) binding events that orchestrate gene expression [13]. Open chromatin regions represent complementary information to gene expression that allows identifying TF and its targets, which are typically missed by transcriptome assays, given its low expression [14]. TF activity signatures, also known as TF footprinting, can be traced back to the identity of the cell [15,16], making it a valuable resource for characterizing tissue heterogeneity. Combining single-cell level data from both gene expression and open chromatin allows us to define sub-populations at higher resolution and study the complex interactions between the genes and the TFs that regulate them. The 10x Genomics Chromium single nucleus snATAC + snRNA platform allows us to profile both open chromatin using the assay for transposase-accessible chromatin (ATAC) [17] and gene expression (GEX) information from the same nuclei, bypassing the need for computational integration of this information retrieved from non-paired cells.

In this study, we characterize experimental protocol optimization for the snATAC + snRNA assay that has been utilized in our laboratory in a variety of studies, including mouse mammary stem cells [18], patient-derived xenografts (PDX) and patient samples from ovarian cancer, and blood samples from COVID-19 patients [19]. Performing simultaneous snATAC and snRNA-seq on the same cells requires careful sample preparation and sequencing involving several steps that will be discussed here: cell isolation (single-cell suspension from the tissue or sample of interest using appropriate methods, such as enzymatic digestion or mechanical dissociation); nuclei isolation (separate the nuclei from the cells using appropriate methods, such as lysis and centrifugation); data quality analysis (analyze the sequencing data to perform integrated analysis of snATAC-seq and snRNA-seq data using appropriate bioinformatics tools).

In contrast to the snATAC + snRNA assay that captures only nuclear transcripts (snRNA), scRNA-seq is an assay that captures transcripts from the whole cell (scRNA), which includes both the nucleus and the cytoplasm. Each method has specific advantages and disadvantages. While scRNA provides a complete transcriptomic profile about individual cells and allows for better identification of rare cell types, the requirement for enzymatic dissociation of samples which may disrupt gene expression profiles, and the time-sensitive nature of the technique makes obtaining high-quality RNA from individual cells challenging [20]. snRNA, on the other hand, can provide more consistent and reliable data since nuclei are easier to isolate and RNA is more stable inside of nuclei than in whole cells [21]. This is especially useful when working with tissues that cannot be easily dissociated [22,23]. The ability to work with frozen samples has also made snRNA a promising tool for the characterization of tissue atlases [24,25]. A disadvantage of this approach is that using nuclei instead of whole cells can result in the loss of information about cell-cell interactions and other critical cellular processes occurring outside the nucleus [26]. In order to compare nuclear transcriptomic signatures against whole cell transcriptomes and their ability to capture the different cell subpopulations we performed scRNA-seq from the same samples that we processed with Protocol A (for snATAC + snRNA assays) in order to have matched samples originating from the same individuals. On analyzing the data obtained from both assays, we compare and characterize the power of each method in identifying immune cell subpopulations and their corresponding gene markers.

## 2. Materials and Methods

### 2.1. Single Nuclei Isolation

Unless alternatively stated, the composition of all the buffers used for all the protocols described was as stated in the cited 10x genomics demonstrated protocols. A schematic of the nuclei isolation protocols used is summarized in Figure 2 (Figure A1 in Appendix A).

### 2.2. Protocol A

Frozen PBMC collected for the investigation of inflammatory cell profiles in patients diagnosed with symptomatic COVID-19 (*n* = 6), clonal hematopoiesis of indeterminate potential (CHIP) (*n* = 6), and with both CHIP and COVID-19 (*n* = 6) were thawed in a 37 °C water bath for 3 to 5 min until no ice was visible. Cells were washed twice with 1 mL phosphate-buffered saline (PBS) + 0.04% bovine serum albumin (BSA) (MilliporeSigma, Burlington, MA, USA) and pelleted at 300 g for 5 min at 4 °C. Dead cells were removed according to the 10x Genomics Protocol [27]. Briefly, using magnetic activated cell sorting (MACS) Dead Cell Removal Kit (Miltenyi Biotec, Bergisch Gladbach, Germany), the pellet was resuspended in 100 mL Dead Cell Removal MicroBeads and incubated for 15 min at room temperature. After incubation, the cell suspension was diluted with 1X Binding Buffer and applied to an MS column. The dead cells were retained in the column, and the live cells passed through the column and were collected. After dead cell removal, the samples were washed twice with 1 mL PBS + 0.04% BSA. The cell concentration and viability were determined using acridine orange propidium iodide (AOPI) Staining solution (VitaStain, Nexcelom Biosciences, Lawrence, MA, USA) and a Cellometer K2 cell counter (Nexcelom Biosciences, Lawrence, MA, USA). The nuclei isolation was performed following the 10x genomics suggested protocol [28]: About $1\times 10^{6}$ cells were pelleted in a 2 mL microcentrifuge tube at 300× *g* for 5 min at 4 °C and resuspended in 100 μL chilled Lysis Buffer by pipetting 10 times. The cells were then incubated on ice for 3 min and, after the addition of 1 mL of wash buffer, they were centrifuged at 500× *g* for 5 min at 4 °C. The wash step was repeated one more time for a total of 2 washes. The pellet was resuspended in a chilled diluted Nuclei Buffer, and the nuclei concentration was assessed by propidium iodide (PI) staining (VitaStain, Nexcelom Biosciences, Lawrence, MA, USA) using a Cellometer K2 cell counter. The quality of nuclei preparation was assessed visually (Figure 3) after observation with an Eclipse 50i microscope (Nikon Instruments, Melville, NY, USA) after staining with 0.4% trypan blue (Gibco, Thermo Fisher Scientific, Waltham, MA, USA). Between 1000 to 5000 nuclei were targeted for capture and used for single nuclei snATAC + snRNA [29].

### 2.3. Protocol B

Fresh tumor tissue was obtained from ovarian cancer patients after debulking surgery. Tissue was transferred in a sterile 10 cm culture dish and finely minced using scissors in a KREBS-ringer bicarbonate (KRB) buffer (20 mM sodium chloride, 5 mM potassium chloride, 25 mM sodium bicarbonate, 20 mM HEPES, and 5.5 mM D-glucose). The sample was centrifuged at 500× *g* for 5 min at 4 °C, and the supernatant was gently removed without disrupting the pellet. Washes were repeated until the supernatant appeared clear. The final pellet was resuspended in 5 mL of KRB buffer with the addition of 2.5 mg/mL of Collagenase IV (MilliporeSigma, Burlington, MA, USA) and Antibiotic/Antimycotic (Gibco, Thermo Fisher Scientific, Waltham, MA, USA), and the tissue was digested in an orbital shaker (85–90 rpm) at 37 °C for 1 h. Following incubation, the dissociated solution was mixed by pipetting up and down 5 times, then strained using 100 μm cell strainer to eliminate bigger aggregates. The flowthrough was centrifuged at 500× *g* for 5 min at 4 °C. The pellet was further digested with 0.05% trypsin-EDTA (Gibco, Thermo Fisher Scientific, Waltham, MA, USA) at 37 °C for 5 min. After neutralizing the reaction with 4 mL complete medium (DMEM [Gibco, Thermo Fisher Scientific, Waltham, MA, USA], 10% fetal bovine serum (FBS) [MilliporeSigma, Burlington, MA, USA], 1% Antibiotic/Antimycotic), the cells were centrifuged at 500× *g* for 5 min at 4 °C, and the pellet was resuspended in 1 mL of PBS + 0.04% BSA. The cell concentration and viability were determined using AOPI Staining solution and a Cellometer K2 cell counter. About $10^{6}$ cells were used for nuclei isolation following the 10x genomics protocol [28]. Followed by targeting 5 thousand nuclei for capture and used for single nuclei snATAC + snRNA [29].

### 2.4. Protocol C

Fresh tumor tissue was obtained from ovarian cancer patients after debulking surgery. The nuclei isolation in protocol C was performed based on the 10x genomics suggested protocol for Complex Tissues for Single-cell snATAC + snRNA [30]. After mincing, a small piece of tissue the size of rice grain was transferred to a 2 mL microcentrifuge tube containing 300 μL of NP-40 Lysis Buffer and homogenized on ice using a pellet pestle. After adding 1 mL of NP-40 Lysis Buffer, the samples were incubated on ice for either 5 or 30 min. After incubation, the samples were strained using a 70 μm cell strainer, and the flowthrough was then centrifuged 500× *g* for 5 min at 4 °C. Nuclei isolation was achieved by resuspending the pellet in d 1 mL PBS + 1% BSA + 1U/μL RNase Inhibitor (Roche, Basel, Switzerland) and incubation on ice for 5 min. After centrifugation at 500× *g* for 5 min at 4 °C, the nuclei were permeabilized by resuspending in 100 μL of 0.1X Lysis Buffer and incubating on ice for 2 min. After adding 1 mL of Wash Buffer, the nuclei were centrifuged at 500× *g* for 5 min at 4 °C, and the pellet was resuspended in Diluted nuclei buffer. The nuclei concentration was assessed by PI staining using a Cellometer K2 cell counter. Five thousand nuclei were targeted for capture and used for single nuclei snATAC + snRNA [29].

### 2.5. Protocol D

Fresh tumor tissue obtained from ovarian cancer patients after debulking surgery was transferred in a sterile 10 cm culture dish and finely minced using scissors in KRB buffer. The sample was centrifuged at 500× *g* for 5 min at 4 °C, and the supernatant was gently removed without disrupting the pellet. Washes were repeated until the supernatant appeared clear. After the last centrifugation, tissue was aliquoted in cryovials and frozen in liquid nitrogen. On the day of the experiment, frozen tissue was cut into small pieces without thawing and used for nuclei isolation, as described in the standad 10x genomics protocol [30]. Five or ten thousand nuclei were targeted for capture and used for single nuclei snATAC + snRNA performed as described in [29].

### 2.6. Protocol E

Tumor tissue obtained from ovarian cancer patients after debulking surgery was frozen in liquid nitrogen as described in protocol D. The day of the experiment, frozen tissue was cut into small pieces without thawing and transferred to a 2 mL microcentrifuge tube containing 300 μL of NP-40 Lysis Buffer and homogenized on ice by using a pellet pestle. After adding 1 mL of NP-40 Lysis Buffer, the samples were incubated on ice for 10 min. After incubation, the samples were strained using a 70 μm cell strainer, and the flowthrough was then centrifuged at 500× *g* for 5 min at 4 °C. Nuclei isolation was achieved by resuspending the pellet in d 1 mL PBS + 1% BSA + 1U/μL RNase Inhibitor (Roche, Basel, Switzerland) and incubation on ice for 5 min. After centrifugation at 500× *g* for 5 min at 4 °C the nuclei were resuspended in 1 mL of PBS + 1% BSA + 1U/μL RNase Inhibitor supplemented with 10 of 7-aminoactinomycin D (7-AAD) ready-made solution (MilliporeSigma, Burlington, MA, USA) and incubated on ice for 5 min. After dissociation, single-nuclei suspensions were isolated into a 5 mL tube via Fluorescence Activated Cell Sorting (FACS) using FACSMelody (BD Biosciences, Franklin Lakes, NJ, USA). Single nuclei were selected with forward scatter, and debris was excluded by size. A negative sample was loaded to determine the 7-AAD positive channel. Dye concentration and sorting buffer composition were as recommended by 10x genomics [30]. A total of 1,152,000 events were processed, and 801,344 positive nuclei were retrieved, with a flow rate of 5, sorting for purity. The viable FACS sorted nuclei were then centrifuged at 500× *g* for 5 min at 4 °C and permeabilized by resuspending in 100 μL of 0.1X Lysis Buffer and incubating on ice for 2 min. After adding 1 mL of Wash Buffer, the nuclei were centrifuged at 500× *g* for 5 min at 4 °C, and the pellet was resuspended in diluted Nuclei Buffer. The nuclei concentration was assessed by PI staining using a Cellometer K2 cell counter. Five thousand nuclei were targeted for capture and used for single nuclei snATAC + snRNA performed as described in [29].

### 2.7. Single Nuclei snATAC + snRNA-Seq

Between 1000 and 5000 nuclei per sample were subjected to transposase assays (exposing buffered nuclei to Tn5 transposase) before proceeding to single-cell partitioning into gel beads in emulsion, barcoding, and pre-amplification. ATAC library construction and cDNA, followed by GEX library construction, were done following established 10x Genomics protocols [29]. The libraries concentration was measured using Qubit High Sensitivity assays (Thermo Fisher Scientific, Waltham, MA, USA), and library profiles were assessed in a fragment analyzer (Advanced Analytical Technologies, Ankeny, IA, USA) before sequencing.

### 2.8. Single-Cell RNA-Seq

The cells were first counted and measured for viability using either the Vi-Cell XR Cell Viability Analyzer (Beckman Coulter, Brea, CA, USA) or a basic hemocytometer and light microscope. The cDNA master mix was prepared according to the manufacturer’s instructions using a minimum concentration of 400,000 cells per milliliter per sample. Library constructions were done following established 10x Genomics protocols for Chromium Next GEM Single-cell 3’ Kit [31]. All cDNA pools and resulting libraries were measured using Qubit High Sensitivity assays and Agilent Bioanalyzer High Sensitivity chips (Agilent, Santa Clara, CA, USA).

### 2.9. Sequencing

The snATAC and snRNA libraries and scRNA libraries were sequenced on an HiSeq 4000 or NextSeq 2000 instrument (Illumina, San Diego, CA, USA) before demultiplexing and alignment to the reference genome (GRCh38/hg38). For each PBMC sample we aimed to sequence 150 million read pairs and 200 million reads for the OC samples, with an optimal number of read pairs per cell between 30,000 and 50,000 (Figure A1A).

### 2.10. Data Analysis

#### 2.10.1. Single-Cell snATAC + snRNA

Sequenced reads from the gene expression (GEX) and DNA accessibility (ATAC) droplet libraries of the snATAC + snRNA assay were processed using 10x Genomics Cell Ranger ARC v2.0.0. The reads were aligned to the pre-built human reference genome GRCh38—v2020-A-2.0.0 (3 May 2021) provided by 10x Genomics. Read trimming, alignment, duplicate marking (ATAC), UMI counting (GEX), peak calling (ATAC), and joint cell calling were performed by Cell Ranger. Quality control statistics calculated by Cell Ranger were compiled, analyzed, and plotted using R [32] and ggplot [33]. Downstream processing, including the calculation of fragment length distribution and transcription start site (TSS) enrichment scores, was done using Signac v1.4.0 [34].

To compare single-cell and single nucleus RNA-seq in the PBMC data, GEX counts matrices from all samples were integrated (batch-corrected) using Seurat v4.0.4 [35] integration method. Sample GEX count matrix from each sample was log-normalized, scaled to mean 0 and variance 1, and dimensionality reduced using PCA on the top 2000 variable genes across all samples. Default options (CCA and pairwise anchor-finding) were used in the integration anchor-finding step. Uniform manifold approximation and projection uniform manifold approximation and projections (UMAP) were calculated using principal components 1 to 50. The integrated data included 57,817 cells from 18 samples. Cells with more than 50% of reads mapped to mitochondrial genes, those with less than 200 unique genes detected (GEX), those with less than 200 unique peaks detected for ATAC, and those with TSS enrichment score (as calculated by Signac) less than one were removed. After quality control (QC) filtering, the number of cells was reduced to 43,160. Cell type identification was made using two methods for comparison, first by using SingleR v1.10.0 [36] with the Monaco immune data from celldex v1.6.0 [37] as a reference, and second by using Azimuth algorithm [8] to map the snATAC + snRNA data to the scRNA data generated from the same cohort of patients, then transferring the labels from the scRNA data to the snATAC + snRNA data. Pseudobulk profiles for the principal component analysis were calculated using muscat [38]. Marker genes were identified using the FindAllMarkers function from Seurat which finds all the differentially expressed genes for each of the cell types using Wilcoxon Rank Sum test (*p* < 0.05).

#### 2.10.2. Single-Cell RNA-Seq

Sequenced reads from the droplet libraries were processed using 10x Genomics Cell Ranger v6.0.1 [39]. The reads were aligned to the pre-built human reference transcriptome GRCh38—v2020-A (7 July 2020) provided by 10x Genomics. Read trimming, alignment, UMI counting, and cell calling were performed by Cell Ranger. Doublet prediction on scRNA-seq data was made using Scrublet v0.2.1 [40] with default parameters. Downstream processing was done using Seurat. Count matrices from all samples were integrated using Seurat v4 integration method. The count matrix from each sample was first normalized and dimensionality reduced as described for snATAC + snRNA. The reciprocal PCA and reference-based integration options were applied in the anchor-finding step. Six samples were chosen as references, and PCs 1-50 were used for the reference-based integration. Cells with more than 50% of reads mapped to mitochondrial genes, those with less than 200 unique genes detected, and those that were predicted as doublets by Scrublet were removed, resulting in 70,184 cells. UMAP was made using the top 50 PCs obtained by running PCA on the integrated (batch-corrected) gene expression matrix. Cell type identification was made with SingleR using the Monaco immune data from celldex as a reference.

All computer programs used for analyzing the data, computing statistics, and generating plots are made publicly available as a GitHub repository (accessed on 6 June 2023). (https://github.com/LabFunEpi/Multiome_MDPI_Genes).

## 3. Results

### 3.1. Nuclei Isolation for snATAC + snRNA Sequencing

PBMC samples (*n* = 18) were processed according to protocol A (Figure 2). Blood was derived from patients with symptomatic COVID-19, CHIP, or both (Methods). From cell suspensions, nuclei can be extracted without a dissociation step according to protocol A (Figure 2), facilitating the acquisition of high-quality nuclei (Figure 3A), according to the standard (Figure 3B).

snATAC + snRNA sequencing was also performed on 18 patients diagnosed with ovarian cancer. The tumor tissue was collected from debulking surgery. For solid tumor samples ensuring adequate tissue dissociation poses several challenges, especially with cancer samples, recognized for their complexity and significant inter-patient variability. A suitable protocol has to promote tissue dissociation, preserving nuclei membrane integrity across all the samples.

Tissue digestion status evaluation during sample preparation was critical. The presence of debris, nuclei aggregates, and membrane damage can negatively impact the assay and the sequencing quality. Over or under-digestion is equally detrimental to the data quality. It is important to obtain complete tissue dissociation of tissue and cell aggregates to avoid clogging the microfluidic system during the capture step while maintaining the integrity of the nuclear membrane. Residual peri-nuclear cytoplasm can also reduce the quality of GEX sequencing results.

Several measures can be taken if nuclei suspension contamination is recognized. Utilizing a fine needle syringe was helpful in dissociating nuclei aggregates; if needed, a strainer was used to remove unwanted debris and cell aggregates. Choosing a protocol tailored for sample characteristics minimizes several of these issues. Lastly, optimization of the duration of cell lysis, especially in the case of tissue dissociation, was also critical (see below).

In 9 OC samples, nuclei isolation for single-cells followed collagenase-based tissue digestion (Figure 2; protocol B). The nuclei isolation protocol for complex tissues C, D, and E (Figure 2) was tested in 11 OC samples. Two samples using protocol B were removed from further processing due to low-quality nuclei (Figure 3A; protocol B).

Protocol performance was compared during sample preparation by nuclei morphology, and its implications on downstream data quality are further discussed. Nuclei isolation following protocol B resulted in one sample clogging the microfluidic system. Under microscopic visualization, this sample had numerous debris (tissue remnants and lipid components), the most likely reason for sample clogging (Figure 3A; protocol B). Another specimen was discarded due to the amount of debris. Also, sample 7 (Figure 3A) exemplifies that some nuclei had intact membranes for protocol B, but frequently membrane blebs were noticed. Fewer aggregates were visualized with complex tissue dissociation (protocols C and D, Figure 3A), and no clogging events were observed after incubation time increment. In addition, protocols C and D showed better preservation of nuclear membranes (Figure 3A).

The incubation step for protocols C and D requires sample-based time optimization. Five and thirty minutes incubation times were tested using the same OC sample (sample 8). Both time periods yielded single nuclei suspension and minimal debris or aggregates (Figure 3A, protocol C). Interestingly, the shorter incubation time (5 min) failed the barcoding step. This resulted in less sample volume after the capturing step and uneven mixing compared to the 30 min incubation.

Targeting 5000 cells was considered ideal for both PBMC and OC. However, due to the lower viability of the PBMC samples, 4000 was the highest target number of cells. For OC samples, the majority were targeted at 5000 (*n* = 16). The ratio between the number of cells retrieved and the number of cells targeted, known as capture efficiency, is demonstrated in Figure A2A.

Ovarian cancer samples (*n* = 18, 90%) that met nuclei preparation quality criteria from all protocols and PBMC samples (*n* = 18) were sequenced according to the 10x Genomics Multiome platform (Methods).

### 3.2. Sequencing Data QC for PBMC (n = 18) and Fresh OC Samples (n = 11)

Sequenced reads (Figure A1A) from ATAC and GEX libraries were processed using 10x Genomics Cell Range ARC. Five representative QC statistics from the ATAC libraries and six from the gene expression (GEX) libraries, as reported by Cell Ranger ARC, were considered for quantitative sample quality comparison (Figure 4A). Downstream data analysis was performed using Signac, where QC statistics per cell were analyzed, allowing for filtering high-quality cells, if necessary. Here PBMC and OC samples QC statistics are compared according to their nuclei isolation protocol (Figure 4A–D).

Globally, the PBMC samples performed below the cut-off for ATAC fraction of high-quality reads and ATAC TSS enrichment score (Figure 4A). Cells under inflammatory stress are expected to produce poor-quality results, especially under stressors such as viral infection and CHIP (typically caused by clones carrying mutations in epigenetic regulators like *TET2*, *DNMT3A* and *ASXL1* [41]), contributing to a lower fraction of high-quality reads. Despite that, the mean TSS fragment length distribution was adequate (Figure 4C). In this situation, cell-level quality control analysis and filtering (Figure 4D) are crucial to ensure that only high-quality cells are used to make biological inferences.

Significant heterogeneity among samples on ATAC and GEX components was observed for ovarian cancer samples (Figure 4A), possibly due to the high complexity of cancer tissue components. ATAC fraction of high-quality reads, ATAC TSS enrichment score, GEX median unique molecules identified (UMI) counts per cell, and GEX fraction of transcriptomic reads parameters can guide analysis decision-making since they have a recommended threshold regardless of the specimen.

ATAC fraction of high-quality fragments represents the percentage of high-quality fragments with a valid barcode associated with cell-containing partitions [Technical Note – Cell Ranger ARC Web Summary Files for Single-Cell snATAC + snRNA ATAC + Gene Expression Assay • Rev A]. When this fraction is less than 40%, it indicates that a significant proportion of ATAC fragments were not associated with a barcoded cell. This can be explained by high ambient ATAC contamination, increased cell accessibility, or poor nuclei preparation. All samples processed according to protocol B (*n* = 7) did not meet the minimum threshold (Figure 4A). Remarkably, all samples (*n* = 10) with the nuclei extracted following protocol C had a higher than 40% threshold for ATAC fraction of high-quality fragments (Figure 4A; Protocol C mean = 76.9%; Protocol D mean = 54.4%).

The median UMI count represents gene expression elements (RNA molecules converted into cDNA) per cell, thus correlating with sequencing depth. Given that all the samples had an adequate number of reads (Figure A1A), read depth was further evaluated by the number of unique genes detected per cell, equally consistent between protocols (Figure 4A).

ATAC fragment size distribution also correlates with sample quality. To characterize DNA accessibility, samples must have fragments of the inter-nucleosome region around them. This distribution is only achieved with protocol C for fresh ovarian cancer samples (Figure 4C), suggesting that protocol B is not feasible for fresh cancer samples.

GEX fraction of transcriptomic reads translates to RNA percentage aligned with the genome and assigned to a barcoded cell [Technical Note – Cell Ranger ARC Web Summary Files for Single-Cell snATAC + snRNA ATAC + Gene Expression Assay • Rev A]. Lower values can be due to environmental RNA contaminating the single-nuclei droplet. In accordance, the protocol C sample outperformed protocol B preparation. Considering the apparent superiority trend, a paired comparison was conducted to control for sample specificities.

The complex tissue dissociation protocol was better than protocol B for OC samples in the paired comparison (Figure 4B, sample 7). Protocol C had an ATAC fraction of high-quality reads of 70% compared to 26% obtained by the B protocol. Also, the complex tissue dissociation protocols’ GEX fraction of transcriptomic reads was 2.4 times higher (Figure 4B). As was foreseeable, the quality differences impacted cell identification and clustering (Figure A3A, sample 7). Taken together, optimal nuclei preparation has been proven crucial to assay performance.

To further optimize protocol C, two incubation times were tested. Sample 8 (Figure 4B) was dissociated for 5 and 30 min. As mentioned, the shorter incubation period for protocol C failed the barcode step, resulting in empty droplets that were wrongly counted as cells (20,000 against 5752 for the 30 min incubation), culminating in a poor GEX fraction of transcriptomic reads per cell and a higher percentage of filtered-out cells. These technical issues culminated with inadequate clustering (ATAC and GEX) for sample 8 when dissociated for only five minutes (Figure A3B), showing the same pattern observed in compromised sample examples in 10x genomics technical notes [42].

An intermediary incubation time point was tested to assess for possible tissue over-digestion within 30 min of incubation. Fresh preparation of sample 9 was incubated for 10 and 30 min. Both preparations yielded good QC values, yet the longer incubation had a lower GEX fraction of transcriptomic reads (Figure 4B). As previously mentioned, tissue over-digestion can impact GEX reads, even if membrane changes are not detected (Figure 3A, protocol C); this probably happened in the 30-minute incubation time. Also, the longer incubation had fewer cells retrieved (1937 compared to 3740 for 10 min). However, both targeted five thousand cells and had similar sample concentrations (4850 nuclei/μL for 10 min, against 3520 nuclei/μL), decreasing capture efficiency for the 30 min incubation. Similar clustering was obtained for ATAC and GEX, demonstrating that, regardless of the absolute QC values, both incubation time points were appropriated for the OC preparation (Figure A3C).

### 3.3. Frozen OC Sample Optimization

After optimizing nuclei extraction protocols for fresh tissue, the same process was performed for frozen tissue. The advantages of frozen tissue include allowing retrospective sample studies and curated specimen selection. Sample 9, previously discussed in the fresh sample preparation, was also processed after being frozen for paired comparisons.

The OC frozen preparation of sample 9 was sequenced before and after FACS sorting (protocols D and E, respectively) (Figure 2). The rationale for sorting was to minimize the need for longer incubation and debris removal. This condition was tested only in frozen tissue since it has lower viability and higher nuclear remnants. FACS nuclei were selected based on 7-AAD signal (Figure A1B). A head-to-head comparison of nuclei, QC analysis, and clustering was performed.

Nuclei morphology was satisfactory throughout (Figure 3A). Albeit the nuclei suspension for the FACS samples had no debris (Figure A1C), both samples performed similarly in the barcoding step. Furthermore, all QC values met the standard threshold and were consistent within samples (Figure 4B). Clustering for GEX and ATAC was also similar between both frozen preparations (Figure A3C). Considering that FACS did not improve any of the mentioned parameters, protocol D was favored over protocol E. However, this approach may be suitable for other sample types.

Accounting for all the samples processed according to protocol C (*n* = 5) and protocol D/E (*n* = 6), DNA accessibility and RNA expression are consistent throughout (Figure 4A). Interestingly, frozen samples detected more unique genes per cell, ensuring greater sequencing depth and cell population representation (Figure 4D). These results suggest that frozen tissue recapitulates the quality and biology of fresh tissue. In that manner, frozen tissue preparation became the standard of our laboratory for OC samples.

### 3.4. Comparison of the Transcriptomes Profiled from snATAC + snRNA and scRNA-Seq from a Matched Cohort

By performing both scRNA-seq and snATAC + snRNA assays on matched samples originating from the same individuals, we compared nuclear transcriptomic signatures against whole cell transcriptomes and their ability to capture immune cell subpopulations. scRNA profiles of 70,184 cells and snRNA profiles of 43,160 nuclei from matched PBMC samples from 18 patients were pooled. Dimensionality reduction using principal component analysis (PCA) and uniform manifold approximation and projection (UMAP), followed by cell type identification using SingleR, allowed us to identify the typical immune cell populations of B-cells, natural killer (NK) cells, CD4+ T-cells, CD8+ T-cells, CD14 monocytes, CD16 monocytes, and an intermediate monocytic population among others (Figure 5A). The cell type identification method used here is a reference-based method wherein the transcriptomic profiles of individual cells are mapped to reference profiles of known cell types which in this case come from the Monaco immune data - bulk RNA-seq samples of sorted immune cell populations from GSE107011 [37]. To evaluate the effect of the reference used for cell type identification, we compared two approaches for labeling cell types in the snRNA data - first using the Monaco reference and secondly by mapping the snRNA data onto the scRNA data and then transferring labels using Azimuth. Since the snRNA and scRNA data come from matched samples, the latter approach should be able to make more biologically meaningful mapping of cell states. Both snRNA and scRNA captured a very similar distribution of cell types, with subtle differences between the two cell type identification methods (Figure 5A,B and Figure A4A). We observed a decreased proportion of NK cells in snRNA when using scRNA as a reference compared to the Monaco reference, making the data more in line with scRNA proportions. However, when comparing the distribution of CD4+ T-cells and gamma delta T-cells (gdT), there was an increased proportion of both types in snRNA (using scRNA reference) that differed from both scRNA and snRNA (using Monaco reference). scRNA identified a larger proportion of CD8+ T-cells than snRNA in both approaches.

To better distinguish the two platforms in identifying cell type-specific transcriptomic signatures, we performed PCA on pseudo-bulk profiles obtained by aggregating the counts from each cell type from each platform. The PCA revealed that the first two principal components (PC) cluster the profiles by platform, while the third and the fourth PCs cluster transcriptomic signatures by major cell subpopulations (myeloid and lymphoid lineages) (Figure 5C). This suggests that the batch effect between scRNA and snRNA is captured by the first two PCs, so removing these PCs should facilitate pooled data analysis from both platforms. This effect is consistent regardless of the reference for snRNA cell type identification.

In Figure 5D, the number of cells labeled differently by the two methods of cell type identification were compared. Overall there is a high concordance between the two methods. We observe some cross-labeling of cell types within the lymphoid and the myeloid lineages suggesting that the two methods perform similarly when identifying broad transcriptomic signatures. But they differ in the resolution of subtle signals between cell subtypes. We also compared the number of overlaps between the markers of each cell type—genes that are significantly differentially expressed (*p*-value < 0.05) between cells of each cell type and all other cells across the two platforms (Figure A4B) with snRNA cell types identified using Monaco reference on the left and using scRNA reference on the right. High concordance in marker genes within the lymphoid and the myeloid lineages cell type was observed, reinforcing the previous inference that both cell type identification methods and platforms perform equally in resolving broad cell lineages. At the same time, there are subtle differences in capturing subpopulations.

On analyzing the correlation of gene expression, averaged by cell type and sample, between scRNA and snRNA, the genes were both positively correlated and negatively correlated (Spearman’s correlation test *p* < 0.05) (Figure 5E and Figure A4C). But positively correlated genes were, on average, expressed higher than the negatively correlated genes (Figure 5F). This could mean that genes with a very low detection threshold are recognized with different sensitivity by the two platforms. All the marker genes were positively correlated (Figure 5G), implying that the cell type-specific genes are detected similarly by both scRNA and snRNA.

## 4. Discussion

Performing ATAC and RNA sequencing from the same nuclei increases assay definition and ability to characterize cell states, including the activity of key transcription factors. As described here, it is critical to ensure good nuclei preparations, and different specimens will need specific dissociation steps. For example, nuclei could be directly extracted from PBMC (protocol A), but human tissue samples required a well-defined dissociation process.

Two different tissue digestion methods were tested in OC samples, one based on collagenase and the other using NP-40 as the major detergent suggested by the manufacturer. Collagenase was initially the method of choice to isolate single-cells since it was previously used in ovarian cancer [43]. However, protocol B yielded poor-quality sequencing results, often below the recommended thresholds (Figure 4A). Hence, the NP-40 methodology [30] was assessed in comparison to the collagenase protocol, demonstrating the superiority of the NP-40 method (Figure 4B, sample 7), improvement that positively impacted the clustering quality (Figure A3A, sample 7). NP-40 based protocols (protocols C, D, and E) all outperformed protocol B (Figure 4).Therefore, our method of choice for OC sample processing is Protocol C with 30 min of incubation and Protocol D.

When performing the assay, the only step in evaluating sample quality is through visual assessment of nuclei morphology. According to the standard (Figure 3B), both methods could yield good-quality nuclei when analyzed under the microscope. Although the microfluidic system capture step failed with both protocols (Figure 3A) protocol B and sample 8 protocol C with 5 min incubation, it was more frequent in protocol B, resulting in two samples lost. Furthermore, as previously mentioned, the collagenase-based method (protocol B) had worse QC sequencing values than the NP-40 based method (protocols C, D, and E).

Poor performance in QC correlates with the inability to identify cells, hence poor sub-population characterization (Figure A3). Ensuring that the number of cells obtained is close to the target value (capture efficiency, Figure A2A) is important for the optimal usage of sequenced reads. This loss occurs mainly due to the extensive disposal of reads for improper barcoded nuclei or DNA and RNA fragments arising from contamination. Consequently, fewer cells achieve quality thresholds and can be used for downstream analysis (Figure A2B). This refined protocol optimization is particularly important when dealing with scarce patient samples.

Beyond choosing the appropriate dissociation method, accessing optimal incubation time is needed. Sample 8 exemplifies that for the same specimen, the longer incubation time was decisive for the success of the experiment. Five minutes of incubation, as suggested by 10x genomics [43], was not sufficient to dissociate the tissue (Figure 3A). Consequently, capturing a single-cell suspension and barcoding was not ideal, jeopardizing all the downstream analysis steps (Figure A3B).

The methods yielded fairly different post-sequencing quality control statistics for both ATAC and GEX. All samples from protocol B had their high-quality ATAC fragments lower than recommended threshold. The open chromatin region was adequately assessed when OC samples were processed using the complex tissue dissociation technique (with fresh and frozen tissue, protocols C, D, and E). The previous protocols also exceeded method B in the GEX parameters.

Notwithstanding the importance of Cell Ranger ARC parameters, careful interpretation is needed since sample biology influences the results. For example, the PBMC cohort had inadequate high-quality ATAC fragments per cell, most likely due to the nature of the samples rather than poor sample preparation, emphasizing context-specific interpretation of these scores. In addition, the raw QC statistics do not translate directly to sequencing quality; multifactorial analysis is needed.

The single-cell definition can be used to favor sample quality assessment. All the mentioned criteria can be applied to each individual cell (Figure A3). This separation allows QC filtering by cell, ensuring that only high-quality data is used for downstream analysis.

The snATAC + snRNA platform is at its full potential when used to characterize different cell populations that compose a particular tissue, embryonic development, disease process, or even inter-patient variability. Ensuring adequate preparation for frozen tissue increases assay versatility. Sample 9 paired comparison assured results reproducibility with frozen preparations (Figure A3A). In addition, the high-quality QC values for protocols D and E endorse the feasibility of handling frozen cancer tissues.

Furthermore, the implication of FACS sorting for frozen samples was investigated. Sample 9 was sequenced before and after FACS. There were no significant differences in sequencing QC results (Figure 4B) or sample clustering (Figure A3C), making FACS sorting an optional step. Nevertheless, the goal of sorting in this paper was to increase sample purity. Yet, FACS can be a powerful tool to separate different populations based on cell surface markers or ploidy [44].

Our data conclusively demonstrates that frozen tissue yields quality and cell type identification comparable to fresh tissue, which is consistent with previously published findings [45]. Also, no additional benefits were observed through FACS, when used to enhance sample purity. Moreover, this study supports the suitability of the NP-40 protocol for ovarian cancer tissue applications, emphasizing the importance of longer incubation times, up to six times the manufacturer’s guidelines, depending on tissue specificity.

Lastly, since the snATAC + snRNA platform quantifies gene expression only from within the nucleus, a comparative analysis of data obtained from scRNA and snRNA measurements from matched samples was done to help better inform decisions in study design. Although similar in their output, scRNA, and snRNA approaches vary in identifying different cell types and their respective proportions. We show that adjusting the reference dataset used for the cell type identification, such as using matched scRNA data as a reference for snRNA cell type identification, may further align the two modalities in a biologically meaningful sense. The choice between scRNA and snRNA sequencing should be driven by the biological question, sample composition, and preservation as fresh or frozen tissue.

Each high throughput sequencing strategy has advantages and limitations. scRNA-seq measures the expression profile of the whole cell and preserves an intact membrane allowing for further downstream applications such as CITE-Seq (Figure 1B), which can profile cell surface protein interactions [46]. On the other hand, snRNA-seq is an especially beneficial approach when working with frozen samples, such as those from tissue biobanks and tissue samples that are difficult to dissociate. In a direct comparison of snRNA-seq and scRNA-seq of the adult kidney, Wu et al. showed that snRNA-seq achieves comparable gene detection and spatial information while eliminating dissociation-induced changes in gene expression [47]. Andrews et al. [48] showed that snRNA-seq could identify rare mesenchymal and precursor cell types in the healthy human liver but that certain subtypes of immune cells were only distinguishable in scRNA-seq. While the preparation process of working with scRNA-seq data is a significant source of variability [49], nuclei provide greater RNA stability and have a reproducible expression of transcripts when compared to whole cells [50].

In addition, using nuclei instead of whole cells can result in loss of information about important cellular processes that occur outside the nucleus. While there are differences between the transcripts obtained from the two platforms, our analysis suggests that both platforms equally capture the underlying cell type signatures, and it is possible to pool information from both platforms removing the batch effect by removing the principal components that correlate with the sequencing platforms. Moreover, we observe that highly expressed genes and marker genes are positively correlated between the two platforms. In contrast, genes that are at low detection levels are negatively correlated, suggesting that the two platforms vary in detecting specific low-expression genes. Careful consideration is necessary when choosing the platforms when the genes of interest are lowly expressed. In summary, both scRNA-seq and snRNA-seq are powerful tools for analyzing gene expression, and the choice between them depends on the specific research question and the type of cells being analyzed.

Multiomic single-cell assays are changing biomedical research by integrating diverse omics layers, such as genomic, epigenomic, transcriptomic, and proteomic (Figure 1B). The addition of multiple omics layers on a single-cell allows a greater understanding of cellular processes, contributing to the discovery of novel biomarkers, identification of cell subtypes, and elucidation of molecular networks [4,5]. Furthermore, some of the most recent advances with technologies enabling the integration of spatial information with single-cell resolution are revolutionizing the way we understand tissue organization and interactions [6]. However, further innovation is needed to characterize some of the most important aspects of epigenomic regulation, such as DNA methylation [8]. As the field continues to advance, increasing resolution and incorporating multiplex platforms, multiomic single-cell assays hold immense potential for personalized medicine, drug discovery, and targeted therapies.

In summary, we analyzed how different nuclei preparation can impact the output of snATAC + snRNA sequencing for PBMC and OC samples. To compare methodologies, we utilized several parameters, such as nuclei morphology, sequencing depth, QC values per sample and for each cell, capture efficiency, and dimensionality reduction techniques, such as UMAP. To control for sample specificity, several paired comparisons between protocols were performed. Also, the PBMC cohort had both snRNA and scRNA sequencing results compared for differences in cell identification. More importantly, our study highlights how the protocols and methods used can be very tissue-dependent and researchers should consider the specificities of each tissue when carrying out their own experiments. Our results show how careful selection of an appropriate nuclei isolation protocol is needed for good sequencing results and the importance of context-based analysis of QC results. Finally, snRNA and scRNA results were consistent, especially considering cell specific markers and abundant transcripts.

## Figures and Tables

**Figure 1 genes-14-01245-f001:**
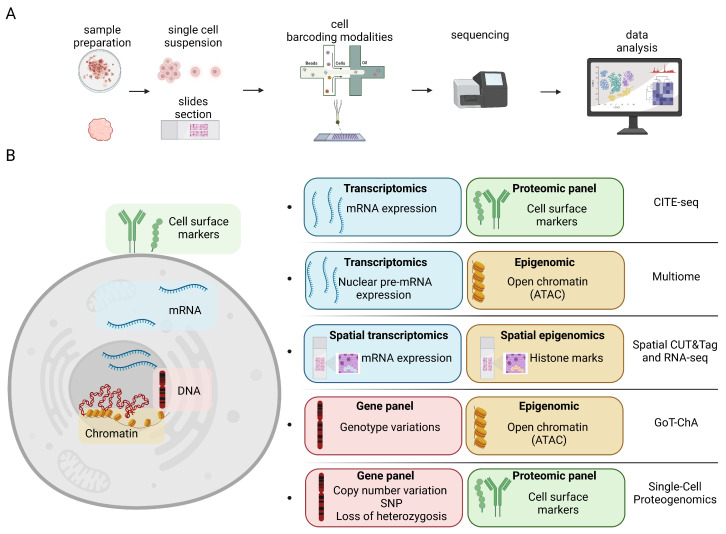
Integrated Single-cell Technologies for Multiomics Approaches. (**A**) Representative pipeline illustrating the sample processing and handling steps. (**B**) Five distinct technologies that integrate two omics layers of information while employing single-cell resolution are illustrated: Cellular indexing of transcriptome and epitopes (CITE-seq), Multiome (snATAC + snRNA), Spatial Cleavage Under Targets and Tagmentation (CUT&Tag) and RNA-seq, Genotyping of Targeted loci with single-cell Chromatin Accessibility (GoT-ChA), and Single-cell Proteogenomics.

**Figure 2 genes-14-01245-f002:**
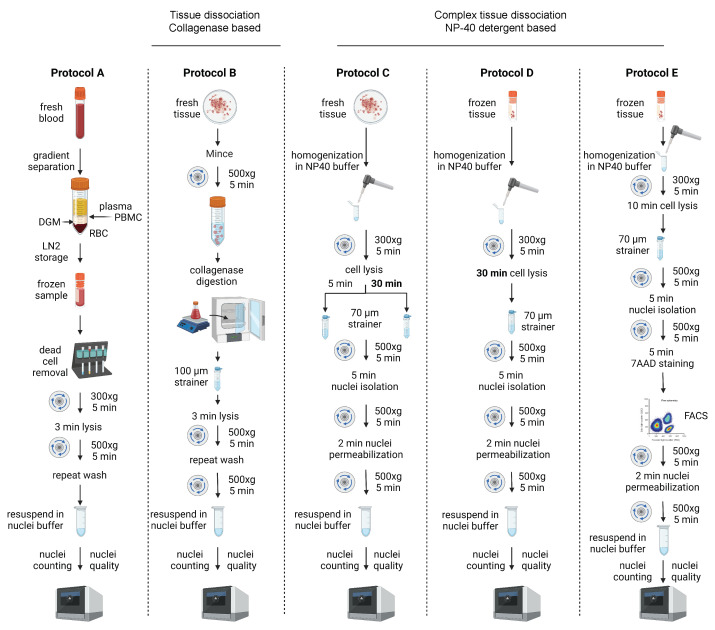
Overview of protocols. Schematic of the main steps pertaining to the five single nuclei isolation protocols discussed.

**Figure 3 genes-14-01245-f003:**
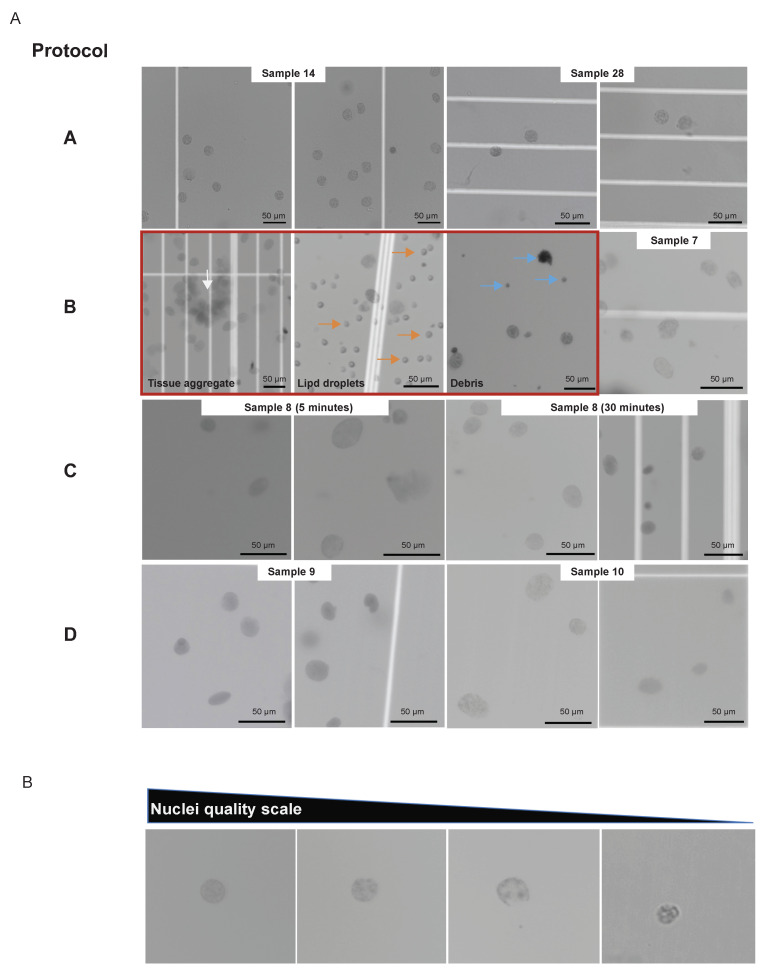
Representative images of single nuclei from the various protocols and samples examined in this study. (**A**) Protocol A shows nuclei isolated from PBMC (samples 14 and 28). Protocol B shows nuclei prepared from ovarian cancer tissues. Shown in order are examples of eliminated samples (inside red square) with undigested tissue aggregates (white arrow), lipid droplet contaminants (orange arrow), and debris (blue arrow), as well as a successful nuclei preparation (sample 7). Protocol C shows nuclei isolated with a 5 min incubation and 30 min incubation from sample 8. Protocol D shows nuclei preparation from samples 9 and 10. (**B**) Representative images of single nuclei organized in decreasing order of preparation quality were used as a standard. Scale bars throughout represent 50 μm.

**Figure 4 genes-14-01245-f004:**
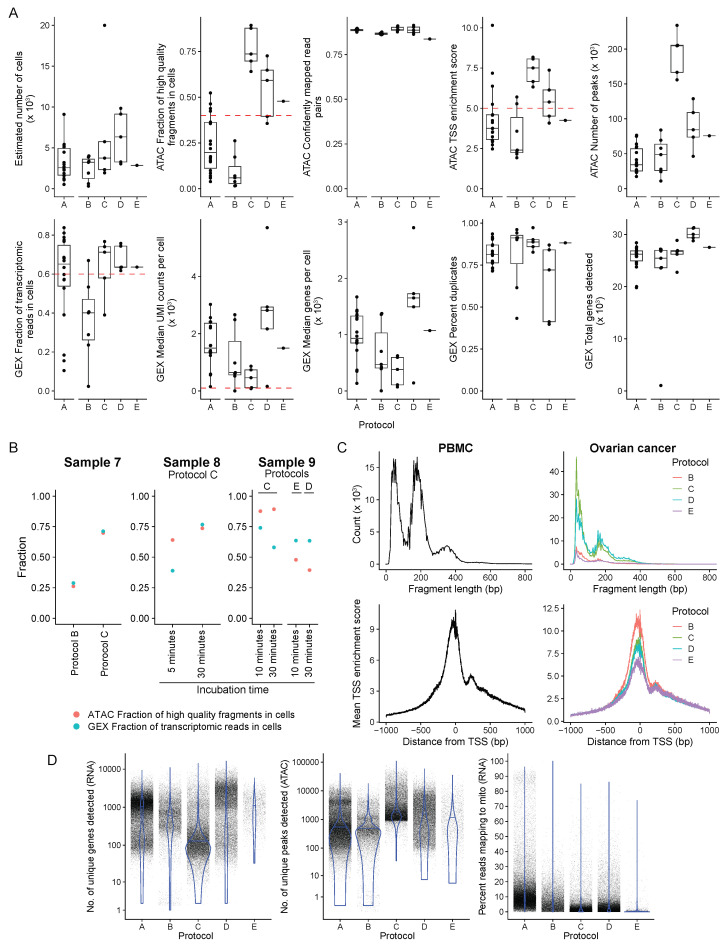
Quality control statistics comparing the protocols examined. (**A**) QC statistics reported by Cell Ranger ARC per sample grouped by protocol, shown as a boxplot where the middle line represents the median, the lower and upper edges of the rectangle represent the first and third quartiles and the lower and upper whiskers represent the interquartile range (IQR) × 1.5. Outliers beyond the end of the whiskers are plotted individually. Protocol A, *n* = 18; Protocol B, *n* = 7, Protocol C, *n* = 5; Protocol D, *n* = 5; Protocol E, *n* = 1. 10x genomics thresholds are represented by red dashed lines. (**B**) The fraction of high-quality fragments in cells (ATAC) and a fraction of transcriptomic reads in cells (GEX) are shown for samples 7, 8, and 9, for which paired protocol and different incubation time comparisons were made. (**C**) Fragment length distribution and mean TSS enrichment score—the ratio of fragments centered at the TSS to fragments in TSS flanking regions from all samples grouped by tissue and protocol. (**D**) Violin plots showing the number of unique genes detected (RNA), number of unique peaks detected (ATAC), and the number of reads mapping to mitochondrial genes (RNA) calculated per cell in all samples and grouped by protocol. Protocol A, *n* = 57,817; Protocol B, *n* = 17,277; Protocol C, *n* = 33,759; Protocol D, *n* = 31,623; Protocol E, *n* = 2853.

**Figure 5 genes-14-01245-f005:**
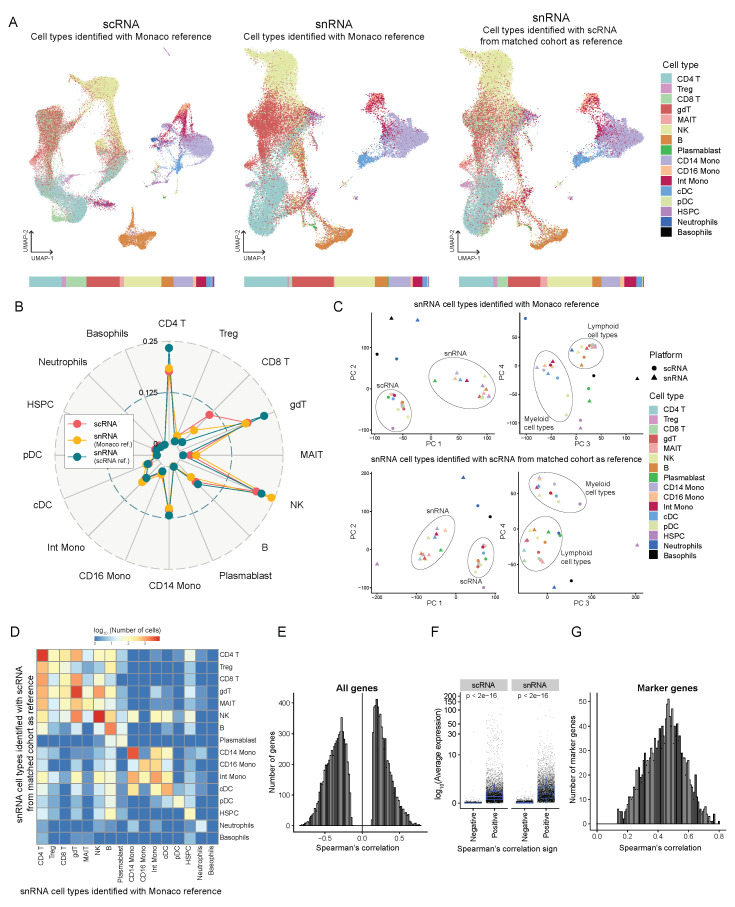
Comparison of the transcriptomes profiled from snATAC + snRNA (GEX) and scRNA-seq from a matched cohort. (**A**) UMAP projection showing 70,184 cells from scRNA colored by cell types identified using SingleR with Monaco reference (left) and 43,160 cells from snRNA, cell types identified using Monaco reference (middle) or using scRNA from the matched cohort as reference (right). The bar at the bottom shows the proportion of each cell type identified using each method. Treg, regulatory T-cells; gdT, gamma delta T-cells; MAIT, Mucosal-associated invariant T cells; NK, natural killer cells; Mono, monocytes; Int, intermediate; cDC, classical dendritic cells; pDC, plasmacytoid dendritic cells; HSPC, hematopoietic stem, and progenitor cells. (**B**) Radar plot showing the comparison of proportions of each cell type identified by each method. (**C**) Principal component analysis of the transcript counts aggregated by platform and cell type with snRNA cell types identified using Monaco reference (top) and scRNA from matched cohorts as reference (bottom). (**D**) Heatmap showing the log (base 10) of the number of cells that overlap cell type identification by the two references—Monaco and scRNA from matched cohort. (**E**) Histogram showing the distribution of genes significantly correlated (Spearman correlation, test for association *p* < 0.05) in expression (averaged over both sample and cell type) between scRNA and snRNA. (**F**) Log (base 10) average expression (averaged over all cells) of genes separated by positive and negative Spearman correlation coefficients. (**G**) Histogram showing the distribution of marker genes significantly correlated (Spearman correlation, test for association *p* < 0.05) in average expression between scRNA and snRNA.

## Data Availability

Data used in this article is available on request.

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
