# Peer review of "Characterization and Optimization of Multiomic Single-Cell Epigenomic Profiling"

_genes, 2023, doi:10.3390/genes14061245_

Round 1

Reviewer 1 Report

Sandoval et al., present a comprehensive evaluation of nuclei extraction for single-cell multi-omic protocols. These types of protocols, and specifically the 10X Genomics Multiome platform which is the focus of the study, are starting to be extensively used in the field. Therefore, this manuscript will be a useful resource for investigators planning single-cell multi-omic experiments in different types of tissues. However, the current version of the manuscript should be revised to improve clarity.

Major revisions:

-       The authors should summarize all protocols discussed through the text (A to E) in Figure 1, rather than presenting some protocol (i.e. Protocol E) in a supplementary figure. Readers should be able to easily visualize all the experimental designs covered. If there’s space constraints in Figure 1, authors could consider dividing protocol schematics in two figures: one related to blood and an additional one related to solid tissue, or one related to fresh and another one related to frozen tissue.

-       The authors should make clear what’s the “standard” ground-truth protocol for solid tissue nuclei dissociation (recommended by Multiome kits), and what are the major differences and improvements between the standard protocol and those presented by the authors.

-       The authors should carefully review their manuscript to make sure that the schematics presented in Figure 1 match the detailed protocol descriptions from Section 2 (Materials and Methods). Currently, there’s several inconsistencies.

-       In Section 3.3, it is unclear if authors would recommend protocol D or E for further experiments (lines 391-396).

Minor:

-       In Figure 3A, the second plot from the left should read “Proportion of confidently mapped ATAC read pairs”

Author Response

We appreciate the time and effort that you and the reviewers have dedicated to providing input to our manuscript. We have been able to incorporate the changes to reflect all the suggestions provided by the reviewers and have highlighted those changes in the manuscript. Here is a point-by-point response to the reviewers’ comments and concerns.

Notes to reviewer 1

Sandoval et al., present a comprehensive evaluation of nuclei extraction for single-cell multi-omic protocols. These types of protocols, and specifically the 10X Genomics Multiome platform which is the focus of the study, are starting to be extensively used in the field. Therefore, this manuscript will be a useful resource for investigators planning single-cell multi-omic experiments in different types of tissues. However, the current version of the manuscript should be revised to improve clarity.

First, we would like to thank the reviewer for pointing out that our work “presents a comprehensive evaluation of nuclei extraction for single-cell multiomic protocols.”

Major revisions:

Comment 1: The authors should summarize all protocols discussed through the text (A to E) in Figure 1, rather than presenting some protocol (i.e. Protocol E) in a supplementary figure. Readers should be able to easily visualize all the experimental designs covered. If there’s space constraints in Figure 1, authors could consider dividing protocol schematics in two figures: one related to blood and an additional one related to solid tissue, or one related to fresh and another one related to frozen tissue.

Response: We thank the reviewer for the valuable suggestion. We added protocol E, to new Figure 2 (previously Figure 1), which includes protocols A through E. We have also moved panel B from the original Figure 1 to Figure S1 to keep the figures consistent and readable.

Comment 2: The authors should make clear what’s the “standard” ground-truth protocol for solid tissue nuclei dissociation (recommended by Multiome kits), and what are the major differences and improvements between the standard protocol and those presented by the authors.

Response: The best nuclei isolation methods for solid tissue are those based on NP-40 as its major detergent as recommended by 10x Genomics, the Multiome kit manufactured. We have now mentioned this in the discussion: “...using NP-40 as the major detergent suggested by the manufacturer” (line 478). Nevertheless, the 5 minutes lysis incubation length recommended in their protocol [Reference 29 of the manuscript] was not sufficient to dissociate solid tumor tissue. Therefore, our main contribution was testing increasing incubation periods and establishing the ideal lysis duration to obtain high-quality nuclei with fewer debris, avoiding sample clogging, and good QC sequencing results. In order to clarify that statement in the text, we included “Therefore, our method of choice for OC sample processing is Protocol C with 30 minutes of incubation and Protocol D”, lines 485-486. 

Comment 3: The authors should carefully review their manuscript to make sure that the schematics presented in Figure 1 match the detailed protocol descriptions from Section 2 (Materials and Methods). Currently, there’s several inconsistencies.

Response: We apologize for the inconsistencies found in our manuscript and thank the reviewer for their careful review regarding inconsistencies. We have thoroughly checked our revised manuscript making sure that it is free of any inconsistencies. One important part of our review process was to include all protocols in the new figure 2 to make readership more accessible.

Comment 4: In Section 3.3, it is unclear if authors would recommend protocol D or E for further experiments (lines 391-396).

Response: We appreciate the reviewer’s comment. For the scope of these experiments, FACS was solely used to augment single nuclei suspension quality, which means avoiding debris and maintaining nuclei membrane integrity while dissociating the tissue. This objective is stated in the original manuscript and in the current one (lines 530-535). To clarify the implications of FACS in the nuclei isolation process and our recommendations regarding its utilization, we have added the following sentence: “Considering that FACS did not improve any of the mentioned parameters, protocol D was favored over protocol E. However, this approach may be suitable for other sample types.” (Lines 404-406).

Minor: 

Comment 5: In Figure 3A, the second plot from the left should read “Proportion of confidently mapped ATAC read pairs” 

Response: We have fixed the new Figure 4A (previously Figure 3A) in the revised manuscript, according to the reviewer’s suggestion. 

Reviewer 2 Report

Sandoval et al. have addressed very important questions in their study that are extremely helpful for the field. Deciding how to process samples and understanding what methods work best is imperative because, if not done correctly, can result in costly mistakes. 

The study as as a whole is a great example of best to go about optimising and testing protocols and methods for experiments. However there are some features of the study that can be improved to make it acceptable for publication.

Major comments:
Analysis of cell type identity and composition was done for the PBMC samples with regards to the different methods of single cell versus single nuclei-seq but not for the different nuclei dissociation protocols. It is clear that data has been generated as there are QC statistics outlined in Figure 3. It would be greatly beneficial for the field to know if the differences in these protocols affected the cell types in the tissues. 

In section 3.4 there is some mention of how the authors annotated the data using Monoco reference and Azimuth label transfer, however this is not stated in the methods. It would be useful if the authors could expand on how they did this.

It must be stated somewhere in the discussion that although studies like this one are very helpful for the field, the protocols and methods used can be very tissue dependent and researchers would need to take this into consideration when carrying out their own experiments. 

Minor comments:
Line 116/117 states that 100um strainer was used, however in Figure 1, protocol B shows 70um.

Methods, section 2.7, this sounds like 10x Genomics Multiome kit, if so it would be useful to reference this kit.

Methods, section 2.9, it would be useful to understand what sequencing depth was targeted.  

Author Response

Notes to Reviewer 2

Comments and Suggestions for Authors

Sandoval et al. have addressed very important questions in their study that are extremely helpful for the field. Deciding how to process samples and understanding what methods work best is imperative because, if not done correctly, can result in costly mistakes.

The study as a whole is a great example of best to go about optimising and testing protocols and methods for experiments. However there are some features of the study that can be improved to make it acceptable for publication.

We would like to thank the reviewer for highlighting the importance of our study.

Major comments: Analysis of cell type identity and composition was done for the PBMC samples with regards to the different methods of single cell versus single nuclei-seq but not for the different nuclei dissociation protocols. It is clear that data has been generated as there are QC statistics outlined in Figure 3. It would be greatly beneficial for the field to know if the differences in these protocols affected the cell types in the tissues.

We thank the reviewer for the very pertinent comment and suggestion.

Firstly, we would like to point out that old Figure 3 (now Figure 4) only includes data from the Multiome (snATAC + snATAC) platform. The comparison between scRNA and snRNA was done only from samples in Protocol A because those were the only datasets for which we had generated matching scRNA data from the same patient samples, in a total of 18 matching pairs We have now included a sentence that clarifies the datasets that we used: “In order to compare nuclear transcriptomic signatures against whole cell transcriptomes and their ability to capture the different cell subpopulations we performed scRNA-seq from the same samples that we processed with Protocol A (for snATAC + snRNA assays) in order to have matched samples originating from the same individuals.” (Lines 76-80)

Secondly, while we agree with the reviewer that comparing cell-type identification (and composition) across samples prepared using different protocols would benefit the field, we also believe that such comparison would require several paired samples treated with the different protocols. More specifically, the problem of cell-type identification in single cell datasets is a computational challenge actively being addressed by the scientific community. There are several strategies to approach this challenge, from using supervised (reference-based) methods (like SingleR) to unsupervised methods (like graph-based clustering and other deep-learning approaches). Performing a comparison as suggested by the reviewer would mean that we are not only comparing the effect of the nuclei preparation protocol, but also the effect of the cell-type identification methodology that we choose, on the resulting cell-type proportions. In other words, cell-type identification methodology is a confounding factor which is not the scope of this work. Those differences in cell type identification become clear in Figure S4, where we see the result of mapping using different methodologies. Inferring any differences in cell identification from one sample (or two) using two different protocols would not be sufficient to draw any conclusions. Importantly, for the comparison between snRNA and scRNA we have 18 paired samples.

To address the effect of the nuclei isolation protocols on the cell-types in an unbiased way (in terms of cell-type identification methodology), we stick to the simple unsupervised clustering performed by Cell Ranger ARC as shown in Figure S3. 

In section 3.4 there is some mention of how the authors annotated the data using Monoco reference and Azimuth label transfer, however this is not stated in the methods. It would be useful if the authors could expand on how they did this.

The cell type identification/annotation methodology is explained in the Methods (sections 2.10.1) in lines 231-236. The annotation and label-transfer steps were done exactly as described in SingleR and Azimuth documentations, which removes the need to expand the steps further.

However, to make it easier for the readers to access the data analysis methodology, we’ve made all the analysis scripts used in this study publicly available as a GitHub repository and added the following sentence: “All computer programs used for analyzing the data, computing statistics, and generating plots are made publicly available as a GitHub repository (https://github.com/LabFunEpi/Multiome_MDPI_Genes)” in lines 256-258.

We have also added the label “Monaco” in lines 233 and 255 to make the reference to the immune data used as reference for cell-type annotation for PBMC samples, consistent throughout the paper.

It must be stated somewhere in the discussion that although studies like this one are very helpful for the field, the protocols and methods used can be very tissue dependent and researchers would need to take this into consideration when carrying out their own experiments. 

We have now included the following sentence in the discussion section (lines 596-598):

“More importantly, our study highlights how the protocols and methods used can be very tissue-dependent and researchers should consider the specificities of each tissue when carrying out their own experiments.”

Minor comments: Line 116/117 states that 100um strainer was used, however in Figure 1, protocol B shows 70um.

We thank the reviewer for pointing out that inconsistency and have now changed it in the figure (Old Fig. 1 now Fig. 2) to 100um.

Methods, section 2.7, this sounds like 10x Genomics Multiome kit, if so it would be useful to reference this kit.

As mentioned by the reviewer, section 2.7 of the study covers the procedures carried out using the Multiome kits. The comprehensive list of reagents and protocols can be found in reference 29, which is included in section 2.7 (10x Genomics. Chromium Next GEM Single Cell Multiome ATAC + Gene Expression Reagent Kits User Guide. Document 690 Number CG000338 Rev F, 2022.). In accordance with lines 85-86, it is indicated that unless otherwise specified, the manufacturer's protocol was adhered to. Given that no optimization was undertaken in the steps outlined in section 2.7, we believe that directing readers to the manufacturer's protocol will provide sufficient information for those interested in replicating the experiment.

Methods, section 2.9, it would be useful to understand what sequencing depth was targeted.  

We have now included in Section 2.9 the sequencing depth that was planned for each sample and refer to figure S1A that includes the actual sequencing depth for all the samples: “For each PBMC sample we aimed to sequence 150 million read pairs and 200 million reads for the OC samples, with an optimal number of read pairs per cell between 30,000 and 50,000.” (lines 206-208). We have also moved two panels from Figure 4 to Figure S1 to have both sequencing depth graphs together, i.e., depth per sample and per cell.

Reviewer 3 Report

This work deals epigenomic-based profilisation of cell-core biological material such as chromatin and expression of genes.

The approach used for such a purpose follows the precise and high-quality extraction/isolation of nuclei material from a single cell and using sufficiently distinguishing analysis on a single-nuclei level.

The morphology of nuclei and NK sequencing were used as ultimate readouts to evaluate efficiency of the isolation method used on PBMCs and OC solid tumour samples.

I value especially the details of some figures, providing easily comprehendible description of the experimental workflows.

Remarks:

1. Some important citations in this area of investigation are missing, in the Introduction section. Please, follow the down below upgrade:

Single-cell (sc) biology is a rapidly evolving field that interrogates various ‘omics

layers (genomics, epigenomics, proteomics) at SC resolution to define cellular heterogeneity in homeostasis and disease [13, https://doi.org/10.1002/pmic.202100198, https://doi.org/10.1016/j.bbamcr.2022.119266].”

These include the type of tissue (PBMC, tumor, bone marrow, or cell line) which can impact the nuclear isolation protocol; the biological condition being studied (homeostasis, development, cancer, infection, or inflammation); the method of sample collection, handling, storage, and maintenance; cell viability and the number of cells available. Moreover, most protocols are typically defined using ideal samples that do not reflect the clinical and translational research environment, the so-called “real world” samples [https://doi.org/10.1002/pmic.202200026].“

2. I would welcome to depict the figure referring to introduction´s statement that about the importance of genomics, epigenomics, proteomics in SC analysis. Please, draw the pipelines of these approaches to see comparisons for the reader.

3. In conclusion section, I would welcome to mention what is the future trend in SC nuclei analysis i.e. what else could be done resp. what are author´s aims is this scope of investigation.

Author Response

Notes to Reviewer 3

Comments and Suggestions for Authors

This work deals epigenomic-based profilisation of cell-core biological material such as chromatin and expression of genes.

The approach used for such a purpose follows the precise and high-quality extraction/isolation of nuclei material from a single cell and using sufficiently distinguishing analysis on a single-nuclei level.

The morphology of nuclei and NK sequencing were used as ultimate readouts to evaluate efficiency of the isolation method used on PBMCs and OC solid tumour samples.

I value especially the details of some figures, providing easily comprehensible description of the experimental workflows.

We would like to thank the reviewer for highlighting the easily understandable descriptions we have provided in our figures for the experimental workflows.

Remarks:

Comment 1: Some important citations in this area of investigation are missing, in the Introduction section. Please, follow the down below upgrade:

“Single-cell (sc) biology is a rapidly evolving field that interrogates various ‘omics

layers (genomics, epigenomics, proteomics) at SC resolution to define cellular heterogeneity in homeostasis and disease [1–3, https://doi.org/10.1002/pmic.202100198, https://doi.org/10.1016/j.bbamcr.2022.119266].”

“These include the type of tissue (PBMC, tumor, bone marrow, or cell line) which can impact the nuclear isolation protocol; the biological condition being studied (homeostasis, development, cancer, infection, or inflammation); the method of sample collection, handling, storage, and maintenance; cell viability and the number of cells available. Moreover, most protocols are typically defined using ideal samples that do not reflect the clinical and translational research environment, the so-called “real world” samples [https://doi.org/10.1002/pmic.202200026].

Response: We greatly appreciate the references suggested by the reviewer, which we agree are indeed essential citations. We have added them in line 25, referencing the following: “Integrating these diverse layers of molecular data allows for a comprehensive analysis of the intricate interplay between genotype...” and referencing the “real world sample” (line 37-38).

Comment 2:  I would welcome to depict the figure referring to introduction´s statement that about the importance of genomics, epigenomics, proteomics in SC analysis. Please, draw the pipelines of these approaches to see comparisons for the reader.

Response: We have introduced a new Figure 1 that illustrates the overall workflow of a multiomic assay in panel A, as well as the key new technologies that incorporate two layers of data, shown in panel B. This figure provides readers with a comprehensive understanding of the information acquired by each multiomic assay and the molecular data captured by each layer. Moreover, this was a great opportunity to highlight novel technologies, which correlates to the reviewer's next comment.

Comment 3:  In conclusion section, I would welcome to mention what is the future trend in SC nuclei analysis i.e. what else could be done resp. what are author´s aims is this scope of investigation.

Response: We greatly appreciate this suggestion to expand on the future of single-cell technology. To summarize the current perspectives in the field, we added the following paragraph with respective references: “Multiomic single-cell assays are changing biomedical research by integrating diverse 'omics layers, such as genomic, epigenomic, transcriptomic, and proteomic. The addition of multiple 'omics layers on a single-cell allows a greater understanding of cellular processes, contributing to the discovery of novel biomarkers, identification of cell subtypes, and elucidation of molecular networks. Furthermore, some of the most recent advances with technologies enabling the integration of spatial information with single-cell resolution are revolutionizing the way we understand tissue organization and interactions. However, further innovation is needed to characterize some of the most important aspects of epigenomic regulation, such as DNA methylation. As the field continues to advance, increasing resolution and incorporating multiplex platforms, multiomic single-cell assays hold immense potential for personalized medicine, drug discovery, and targeted therapies.” (Lines 579-589). 

Round 2

Reviewer 3 Report

The authors have reacted to the given queries.